# Continuous Bioproduction of Alginate Bacterial under Nitrogen Fixation and Nonfixation Conditions

Pablo Contreras-Abara [1], Tania Castillo [2], Belén Ponce [1], Viviana Urtuvia [1], Carlos Peña [2] and Alvaro Díaz-Barrera [1,*]

1 Escuela de Ingeniería Bioquímica, Pontificia Universidad Católica de Valparaíso, Valparaíso 2340000, Chile
2 Departamento de Ingeniería Celular y Biocatálisis, Instituto de Biotecnología, Universidad Nacional Autónoma de México, Ave. Universidad 2001, Col. Chamilpa Cuernavaca, Cuernavaca 62210, Morelos, Mexico
* Correspondence: alvaro.diaz@pucv.cl

**Abstract:** Alginate is a biomaterial produced by *Azotobacter vinelandii*, a diazotroph that, under nitrogen-fixing conditions, can fix nitrogen under high oxygen levels. In *A. vinelandii*, alginate is synthesized from fructose-6P via synthesis of precursor, polymerization, and modification/exportation. Due to its viscosifying, gelling, and thickening characteristics, alginate is widely used in food, pharmaceutical, and cosmetical industries. This study aimed to develop a continuous bioprocess and a comparative analysis of alginate production under diazotrophic and nondiazotrophic conditions. Continuous cultures were developed at three dilution rates (0.06, 0.08 and 0.10 $h^{-1}$). In steady state, the respiratory activity, alginate production, alginate molecular weight and the genes encoding alginate polymerase were determined. Under the conditions studied, the specific oxygen uptake rate and respiratory quotient were similar. The diazotrophic conditions improved the conversion of sucrose to alginate and the specific productivity rate, which was $0.24 \pm 0.03$ g g$^{-1}$ h$^{-1}$. A higher alginate molecular weight ($725 \pm 20$ kDa) was also achieved under diazotrophic conditions, which can be explained by an increase in the gene expression of genes *alg8* and *alg44* (encoding polymerase). The results of this work show the feasibility of enhancing alginate production (yields and specific productivity rates) and quality (molecular weight) under nitrogen-fixing conditions, opening the possibility of developing a continuous bioprocess to produce alginate with specific characteristics under conditions of diazotrophy.

**Keywords:** alginate; mean molecular weight; diazotrophy; chemostat

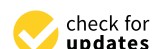



## 1. Introduction

Alginates are polysaccharides formed by monomers of β-D-mannuronic acid (M) and its epimer α-L-guluronic acid (G). Alginate applicability for practical biomaterials has been proven in applications such as hydrogels for three-dimensional extrusion bioprinting [1], due to their ability to form gels. *Azotobacter vinelandii* is a bacterium able to produce polymers such as poly-3-hydroxybutyrate (P3HB) and alginate during cellular growth [2]. Alginate monomers can be arranged in chains of different molecular weights and distributed in MMMM, GGGG, and MGMG blocks [3].

Alginates are also produced by brown algae, but their production presents several disadvantages, which may limit their industrial use. Algal alginates are complex polymer mixtures with a wide range of molecular weights and compositions of M and G residues. Thus, alginates with a well-defined composition cannot be obtained. Current information about the costs of alginate production is not readily available; it is estimated that commercial alginates used in the food and cosmetic industries can be acquired at prices below 5 USD per kilogram. From a commercial standpoint, the most important characteristics of alginates are their viscosity in solutions and their capacity as gelling agents. The aforementioned alginate properties depend largely on the relative content of the two monomers (G and M), the

degree of acetylation, as well as the molecular weight of the polymer [4]. The manipulation of the molecular weight and its distribution can improve the physical properties of resultant gels [5]. One strategy to produce alginates with varied and reproducible physicochemical characteristics is through the manipulation of culture conditions during the bioprocess.

The biosynthesis of alginate in *A. vinelandii* involves three enzymatic stages: synthesis of precursor, polymerization, and modification/secretion [6]. The alginate polymerase complex is composed of glycosyl-transferase/polymerase (Alg8) and copolymerase (Alg44), which are encoded by the genes *alg8* and *alg44* [6]. Díaz-Barrera et al. [7] observed an increase in the relative expression of *alg8* in chemostat cultivations of *A. vinelandii* at D = 0.1 h$^{-1}$ and a correlation between the mean molecular weight and the gene expression of *alg8*, while in their later work [8] conducted under chemostat cultivations at D = 0.07 h$^{-1}$, the alginate molecular weight synthesized in those cultivations did not correlate with a high relative expression of *alg8* or *alg44*, which can be indicative of a balanced transcription of alginate polymerization genes.

*A. vinelandii* is a bacterium able to fix atmospheric nitrogen, a reaction catalyzed by the enzyme nitrogenase, which converts a molecule of $N_2$ into two molecules of ammonia [9]. If the cellular growth is developed in the absence of a fixed nitrogen source (e.g., without ammonium) the culture is performed under diazotrophic conditions, whereas if the culture medium is supplemented with a fixed nitrogen source, the growth condition is nondiazotrophic. Although the enzyme nitrogenase is highly sensitive to oxygen, *A. vinelandii* has a greater capacity to fix nitrogen because of its ability to fix nitrogen even at high oxygen levels [9]. This capability has been related to a higher respiration rate when this bacterium is grown under diazotrophic conditions and high oxygen levels [10]. The effect of oxygen availability in diazotrophic cultivations on the biosynthesis of alginate has been studied by several authors [11–17]. A modality adequate to evaluate alginate production is the use of continuous cultures, in particular, chemostat cultures, in which the cells can be cultivated at a constant specific growth rate (which can be established by the different dilution rates) which allows for distinguishing between the effect of a specific nutrient such as ammonium or oxygen, as well as for monitoring the specific growth rates [8,11].

Based on genome-scale modeling, Tec-Campos et al. [18] described that under diazotrophic conditions, the carbon and nitrogen fluxes through *A. vinelandii* could decrease as a response to nitrogen limitation, whereas when ammonium is supplemented, the cell growth rate and alginate production rate increase. Although there are studies in the literature about alginate synthesis by *A. vinelandii* under diazotrophic and nondiazotrophic conditions [7,14,19], those works are not comparable due to differences in their operating conditions. For this reason, the present study aimed to develop a continuous bioprocess to produce alginate with different molecular weights using *A. vinelandii* in cultures under diazotrophic and nondiazotrophic conditions. In this sense, a bioprocess to produce alginate under similar conditions of cultivation, such as culture medium composition, agitation rate, and particularly specific growth rate (in chemostat-mode) is implemented, because only the condition diazotrophic or nondiazotrophic was varied in this study.

## 2. Materials and Methods

### 2.1. Microorganism, Culture Medium, and Inoculum Preparation

*A. vinelandii* ATCC 9046 (wild-type strain) was used for this study. Chemostats were developed under diazotrophic (nitrogen-fixing) and nondiazotrophic (using ammonium in the culture medium) conditions. The diazotrophic conditions were evaluated using a culture medium that contained gaseous nitrogen coming from the air incorporated as a nitrogen source alone. The culture medium for diazotrophic conditions contained (in g L$^{-1}$): sucrose, 20; $K_2HPO_4 \cdot 3H_2O$, 0.2; $CaSO_4 \cdot 2H_2O$, 0.056; NaCl, 0.2; $MgSO_4 \cdot 7H_2O$, 0.2; $Na_2MoO_4 \cdot 2H_2O$, 0.0029; $FeSO_4 \cdot 7H_2O$, 0.027. Solutions of medium were dissolved and autoclaved at 121 °C for 20 min. To avoid precipitation, NaCl, $MgSO_4 \cdot 7H_2O$, $Na_2MoO_4 \cdot 2H_2O$, $FeSO_4 \cdot 7H_2O$, and $CaSO_4 \cdot 2H_2O$ were separated for sterilization, while the rest of the nutrients were dissolved and autoclaved in the bioreactor. For nondiazotrophic conditions,

the culture medium described supplemented with 0.8 g $L^{-1}$ of $(NH_4)_2SO_4$ was used. *A. vinelandii* cells were incubated at 200 rpm and 30 °C in an orbital incubator shaker (Daihan LabTech CO, Namyangju, Kyungki-Do, Republic of Korea) in a 500 mL Erlenmeyer flask with 100 mL of culture medium. After 20 h of cultivation, the bioreactor was inoculated with 10% *v/v* inoculum.

### 2.2. Culture Conditions

Chemostat cultures were conducted in a 3 L bioreactor (Applikon, Schiedam, The Netherlands) with a working volume of 1.5 L. The pH was controlled at 7.0 ± 0.2 using a 2 N NaOH solution via a peristaltic pump coupled to an EZ-2-Control unit. The bioreactor was equipped with two Rushton turbines and aerated at 1.5 L $min^{-1}$. Dissolved oxygen tension (DOT) was measured by polarographic and was not controlled. After 24 h of cellular growth, the fresh culture medium was fed to the bioreactor, and the broth culture was removed from the bioreactor via a continuously operated peristaltic pump (Cole-Parmer, Vernon Hills, IL, USA).

The chemostat was operated at D values of 0.06, 0.08, and 0.10 $h^{-1}$. After at least 3 residence times, 20 mL samples were taken from the reactor at different times. Steady-state conditions were reached once the optical density reached 540 nm and the sucrose concentration had a variation coefficient below 10%. The results shown are the mean value of two independent chemostat runs, and error bars correspond to the range among the replicates.

### 2.3. Analytical Methods

Biomass and alginate concentrations were estimated gravimetrically. A 20 mL sample of culture broth was mixed with 2 mL EDTA (Ethylenediaminetetraacetic acid disodium salt) (0.1 M) and 2 mL NaCl (0.1 M) and then centrifuged at 10,000 rpm (Thermo Scientific SL-16R, Waltham, MA, USA) for 10 min. The pellet of biomass was washed three times using distilled water and then dried at 100 °C to a constant weight. For alginate quantification, 10 mL of supernatant was mixed with cold propan-2-ol in a 3:1 vol ratio. The resultant precipitate was filtered through 0.45 μm Millipore filter paper and dried at 70 °C to a constant weight. The sucrose concentration was determined by the dinitrosalicylic acid (DNS) reagent [20] after acid hydrolysis with HCl. The alginate mean molecular weight (MMW) was determined by gel permeation chromatography (GPC) in an HPLC with a differential refractometer detector (Jasco, Mary's Court Easton, MD, USA), according to Díaz-Barrera et al. [19]. The P3HB content was determined by extracting the P3HB from the cell and hydrolyzing it to crotonic acid, which was measured by an HPLC-UV (Jasco, Mary's Court Easton, MD, USA) system [21]. The ammonium concentration in the supernatant was determined by the phenol-hypochlorite method [22], and the phosphate concentration was determined by an automatic analyzer Random Access Y15 (BioSystem, Barcelona, Spain).

### 2.4. Gene Expression Analysis

The cells were harvested by centrifugation at 10,000 rpm (4 °C) for 4 min (Thermo Scientific SL-16R), and the pellet was washed three to five times. The mRNA was stabilized and protected by adding RNAlater solution (Thermo Fisher Scientific, Waltham, MA, USA) to the biomass pellet and stored at −80 °C for posterior RNA isolation. RNA was isolated using a High Pure RNA Isolation Kit (Roche Life Sciences, Penzberg, Germany) and treated with RNase-free DNase (Roche) according to the fabrication protocol. The RNA was quantified using a BioSpec-nano system (Shimadzu, Kyoto, Japan). cDNA was synthesized using a RevertAid H Minus First Strand cDNA Synthesis Kit (Thermo Fisher Scientific, Waltham, MA, USA) according to the fabrication protocol. Reverse transcription-real-time PCR (RT-qPCR) was carried out with specific primers shown in Table 1.

**Table 1.** Primers designed for gene expression by qPCR.

| Gene | Primers | Gene | Primers |
|------|---------|------|---------|
| *alg8*-**F** | 5′-TGTTGAACCAGCTCTGGAAG-3′ | *alg8*-**R** | 5′-CCTACCCGCTGATCCTCTAC-3′ |
| *alg44*-**F** | 5′-CGACAACTTCACCGAAGGG-3′ | *alg44*-**R** | 5′-CGACAACTTCACCGAAGGG-3′ |
| *gyrA*-**F** | 5′-ACCTGATCACCGAGGAAGAG-3′ | *gyrA*-**R** | 5′-AGGTGCTCGACGTAATCCTC-3′ |

For RT-qPCR, 100 ng of total RNA was reverse transcribed. Real-time PCR was performed in the AriaMx Real-Time PCR system (Agilent Technologies, Santa Clara, CA, USA) using Brilliant II SYBR@ Green QPCR Master Mix (Agilent Technologies, Santa Clara, CA, USA). The samples were initially denatured at 95 °C for 5 min. A 40-cycle amplification and quantification protocol was used for RT-qPCR (95 °C for 15 s, 59 °C for 15 s and 72 °C for 15 s). Melting curve analyses confirmed the amplification of a single product for each primer pair. The results were analyzed using the $2^{-\Delta\Delta CT}$ method [23,24].

Relative gene expression values were normalized using *gyrA* as a housekeeping gene [16] and presented as fold changes in transcription levels of culture sample growth in diazotrophic conditions with respect to the transcript levels of cultured sample growth in nondiazotrophic conditions.

*2.5. Specific Oxygen Uptake Rate and Respiratory Quotient Determination at the Steady State*

The respiratory quotient (RQ) was determined using oxygen transfer rate (OTR) and carbon dioxide transfer rate (CTR) data from the ratio CTR/OTR. Gas analysis was performed by measuring oxygen and carbon dioxide in the exit and inlet gas with a BlueVary gas analyzer (BlueSense, Herten, Germany). The OTR (mmol $L^{-1}$ $h^{-1}$) and CTR (mmol $L^{-1}$ $h^{-1}$) were calculated using a steady-state gas phase balance, according to Equations (1) and (2):

$$\text{OTR} = \frac{C\,F_G^{in}}{V_R V_M}\left(X_{O_2}^{in} - X_{O_2}^{out}\left(\frac{1 - X_{O_2}^{in} - X_{CO_2}^{in}}{1 - X_{O_2}^{out} - X_{CO_2}^{out}}\right)\right) \qquad (1)$$

$$\text{CTR} = \frac{C\,F_G^{in}}{V_R V_M}\left(X_{CO_2}^{in}\left(\frac{1 - X_{O_2}^{in} - X_{CO_2}^{in}}{1 - X_{O_2}^{out} - X_{CO_2}^{out}}\right) - X_{CO_2}^{out}\right) \qquad (2)$$

where C is unit conversion factor (1000), $F_G^{in}$ is the volumetric inlet air flow under standard conditions (L $h^{-1}$); $V_R$ is the working volume (L); $V_M$ is the mol volume of the ideal gas under standard conditions (L $mmol^{-1}$); $X_{O_2}^{in}$ is the molar fraction of oxygen in the inlet air (mol $mol^{-1}$); $X_{O_2}^{out}$ is the molar fraction of oxygen in the outlet fermentation gas of the bioreactor (mol $mol^{-1}$); $X_{CO_2}^{in}$ is the molar fraction of carbon dioxide in the inlet air (mol $mol^{-1}$); and $X_{CO_2}^{out}$ is the molar fraction of carbon dioxide in the outlet fermentation gas of the bioreactor (mol $mol^{-1}$).

During continuous operation in the steady state, $\frac{dC_L}{dt} = 0$ ($C_L$, dissolved oxygen concentration in the liquid) and the OTR is considered to be equal to the oxygen uptake rate (OUR).

*2.6. Carbon Balance*

Carbon distribution to the different steady states evaluated was determined from reactor mass balances. For the analysis, the calculation was performed considering that the carbon source is converted mainly into biomass, P3HB, alginate and $CO_2$ [25].

*2.7. Fermentation Parameters*

The yields of biomass and alginate based on sucrose ($Y_{x/s}$ and $Y_{p/s}$, respectively), the yield of alginate based on biomass ($Y_{p/x}$), specific oxygen uptake rate ($qO_2$), and specific production rate ($q_p$) were calculated at a steady state for each condition, considering the

dilution rate (D; $h^{-1}$), alginate concentration in the steady state (P; g $L^{-1}$), OTR values (mmol $L^{-1}$ $h^{-1}$), biomass concentration (X; g $L^{-1}$) in the steady state, sucrose concentration in the steady state (S; g $L^{-1}$), and sucrose concentration in the feed medium (Sr; g $L^{-1}$), as indicated by the following equations:

$$Y_{x/s} = \frac{X}{(S_r - S)} \tag{3}$$

$$Y_{p/s} = \frac{P}{(S_r - S)} \tag{4}$$

$$Y_{p/x} = \frac{P}{X} \tag{5}$$

$$q_p = \frac{DP}{X} \tag{6}$$

$$qO_2 = \frac{OTR}{X} \tag{7}$$

## 3. Results and Discussion

### 3.1. Biomass, P3HB, and Alginate Concentration in the Steady State under Diazotrophic and Nondiazotrophic Conditions

Figure 1 shows the biomass, alginate, and P3HB concentrations at steady state in the chemostat cultures performed under diazotrophic and nondiazotrophic conditions. In all the conditions evaluated, the biomass concentration was higher under nondiazotrophic conditions (Figure 1a), which could be related to the availability of ammonium to stimulate the production of biomass. In this context, under nondiazotrophic conditions, the energetic requirements to produce cells are lower than those in cultivations in which nitrogen fixation occurs [26,27]. Under the conditions evaluated, low P3HB production was obtained (Figure 1b). Thus, the highest P3HB concentration (0.35 ± 0.05 g $L^{-1}$) and polymer content in the cells (34.3 ± 3.0% w $w^{-1}$) were obtained under diazotrophic conditions at a D of 0.06 $h^{-1}$. In the steady states conducted at a D of 0.08 and 0.10 $h^{-1}$, a low P3HB concentration (less than 0.09 g $L^{-1}$) and P3HB content (less than 8.5 ± 2.0% w $w^{-1}$) were obtained. Similarly, Díaz-Barrera et al. [8] observed that in diazotrophic chemostat cultures, the highest P3HB content was achieved at 0.07 $h^{-1}$, and it dropped at 0.09 $h^{-1}$. These results agree with other studies in which it has been reported that P3HB biosynthesis is mainly enhanced at low specific growth rates [14].

It is known that acetyl-CoA is the acetyl donor for alginate acetylation and is a precursor for P3HB biosynthesis [6]. In this regard, it is possible that under diazotrophic conditions and a low D (in which more P3HB was produced, such as in Figure 1b), a higher proportion of acetyl-CoA could be canalized to synthesize P3HB instead of being used for acetylation of alginate. Other analyses of alginate composition could be carried out to evaluate this possibility.

It is clear from the results that the cultivation conditions were not propitious to produce P3HB. Considering the P3HB content in the cells, the biomass concentration values without including the P3HB content were also highest under nondiazotrophic conditions (Figure 1c).

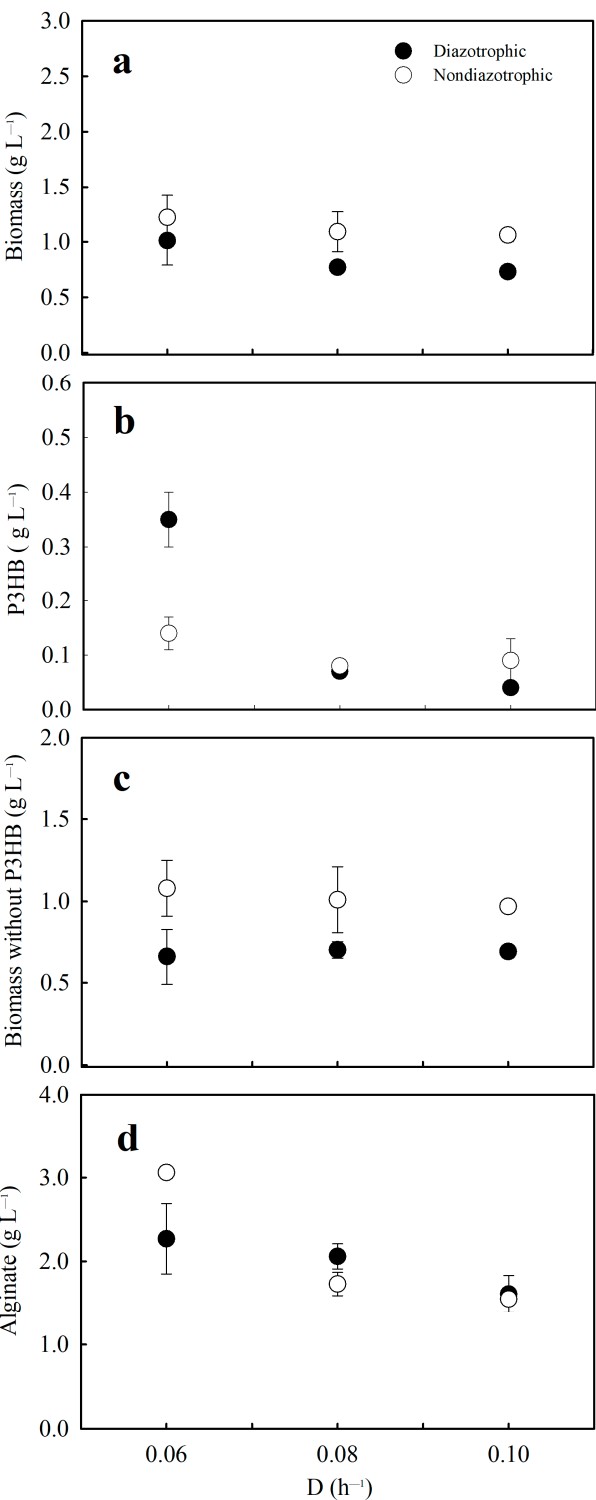

**Figure 1.** Steady-state biomass concentration (**a**), P3HB concentration (**b**), biomass concentration without P3HB (**c**) and alginate concentration (**d**) under diazotrophic (black circles) and nondiazotrophic (white circles) conditions in *A. vinelandii* continuous cultures.

Regarding the alginate concentration in the steady state, independent of the cultivation conditions, the alginate concentration decreased with increasing D (Figure 1d). The highest alginate concentration ($3.06 \pm 0.02$ g L$^{-1}$) was obtained at the lowest D assayed under nondiazotrophic conditions (Figure 1d). In this condition, the steady-state carried out to the higher D ($0.10$ h$^{-1}$) showed the lowest alginate production, which reached

$1.55 \pm 0.02$ g L$^{-1}$. Previously, under noncontrolled oxygen conditions (as in this study), a decrease in the alginate concentration by increasing D was observed [8,28]. However, under DOT control (for example, DOT of 1%), by increasing D (from 0.06 to 0.10 h$^{-1}$), the alginate concentration also increased [19]. From a productive point of view, this evidence is important because to develop a bioprocess without DOT control (and hence develop it in a way that is less expensive), it is necessary to perform cultures with a lower D at the highest residence time, but also at a higher alginate production rate. Chemostat cultures provide nutrient-limiting conditions specific for a single nutrient in a medium with stable levels of the nonlimiting components [29]. In our study, the cultures were not limited by carbon because between 8.1 and 16 g L$^{-1}$ sucrose remained in the bioreactor at a steady state (Table 2). In addition, the ammonium concentration in the steady state was not detected in all the conditions evaluated (data not shown), and phosphate levels under nondiazotrophic conditions were tenfold lower than those reached during diazotrophic conditions (Table 2). In light of this evidence, it is possible that under nondiazotrophic conditions, phosphate could be a nutrient limiting cellular growth.

**Table 2.** Sucrose and phosphate concentrations in steady state during continuous cultures of *A. vinelandii* conducted under diazotrophic and nondiazotrophic conditions.

| D (h$^{-1}$) | Diazotrophy | | Nondiazotrophy | |
|---|---|---|---|---|
| | Residual Sucrose (g L$^{-1}$) | Residual Phosphate (mg L$^{-1}$) | Residual Sucrose (g L$^{-1}$) | Residual Phosphate (mg L$^{-1}$) |
| 0.06 | $14.2 \pm 0.6$ | $38.7 \pm 11.9$ | $8.9 \pm 0.2$ | $3.4 \pm 0.6$ |
| 0.08 | $14.2 \pm 0.5$ | $24.2 \pm 0.4$ | $8.1 \pm 0.5$ | $2.7 \pm 0.9$ |
| 0.10 | $16.1 \pm 0.6$ | $29.6 \pm 5.2$ | $9.0 \pm 3.1$ | $0.7 \pm 0.2$ |

*3.2. qO$_2$ and RQ at Different Dilution Rates under Diazotrophic and Nondiazotrophic Conditions*

In all the steady states evaluated, the DOT was nearly zero (data not shown), which has been previously reported [7]. This condition (DOT~0) is indicative that the steady states were conducted under oxygen limitation. In each culture developed, the qO$_2$ and RQ were evaluated at the steady state in the chemostat cultures (Figure 2). Comparing diazotrophic and nondiazotrophic conditions reveals that the qO$_2$ did not show significant differences at the three D evaluated (Figure 2a), which was not expected. Previous evidence has indicated that a higher qO$_2$ is expected during diazotrophic cultivation as a response to the protection of the nitrogenase complex during nitrogen fixation [9,11].

Nevertheless, recent quantitative mathematical models of nitrogen fixation in *A. vinelandii* have shown that oxygen-scavenging respiration is not a "switch-on/switch-off" mechanism but is performed at different carbon-nitrogen ratios (C/N), even when the cells do not fix nitrogen [10]. Inomura et al. [10] proposed controlling respiration via the C/N ratio, in which excess substrate respiration increases with the C/N ratio until nitrogenase can be derepressed when the C/N ratio is high. It is known that *A. vinelandii* possesses the capacity to fix nitrogen at high oxygen concentrations [30], which depends on different mechanisms to protect nitrogenase from inactivation by oxygen [9]. One mechanism involves respiratory protection, which is based on a high respiration rate to maintain a low oxygen concentration. In light of the evidence, a similar qO$_2$ under diazotrophic and nondiazotrophic cultivation conditions (Figure 2a) could be explained by the excess respiration mechanism remaining active independent of nitrogen-fixing activity. In this regard, Alleman et al. [31], suggested that the respiratory protection mechanism might be a core principle of metabolism using a genome-scale metabolic model.

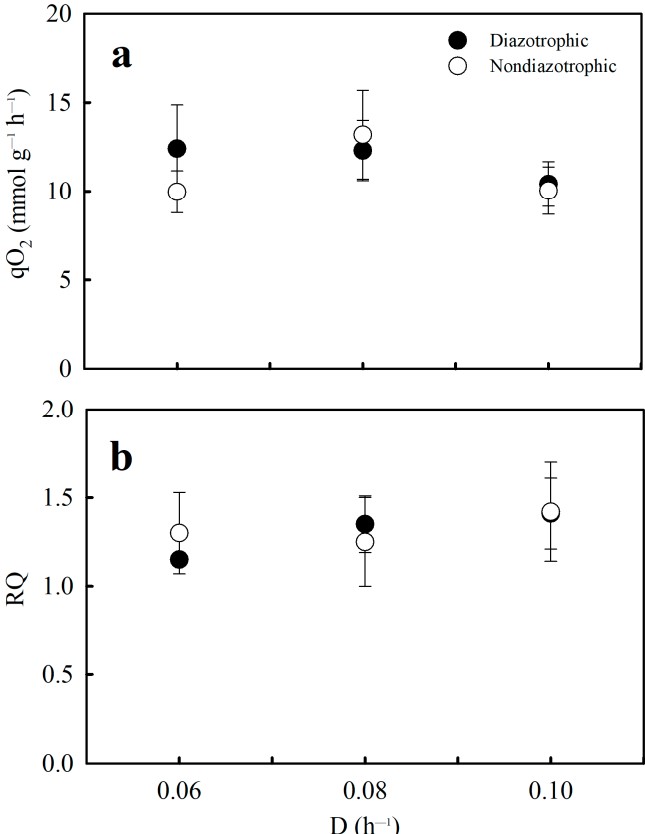

**Figure 2.** The specific oxygen consumption rate (qO$_2$) (**a**) and respiratory quotient (RQ) (**b**) under diazotrophic (black circles) and nondiazotrophic (white circles) conditions in *A. vinelandii* continuous cultures.

Similar RQ values over 1.0 were obtained under the conditions studied, reaching between 1.2 ± 0.02 and 1.42 ± 0.28 (Figure 2b). Similar values were reported by Sabra et al. [28] in oxygen-limiting chemostat cultivations under diazotrophic conditions. Those authors [28] calculated a theoretical RQ value of 0.8 associated with higher conversion of the carbon source to alginate, whereas for PHB synthesis, an RQ value of 1.33 was calculated. In concordance, similar RQ values obtained in our experiments can be related to a similar alginate concentration reached in the steady state under diazotrophic and nondiazotrophic conditions (Figure 1d). Our evidence (RQ above 1.0) shows that it is possible to manipulate culture conditions (for example, increasing the agitation rate and hence OTR) to decrease the RQ during the steady state to improve alginate production. Similarly, Díaz-Barrera et al. [17], in batch cultures of *A. vinelandii*, related a decrease in the RQ from 1.3 to 0.8 with an increase in the alginate concentration from 2.1 to 3.3 g L$^{-1}$.

### 3.3. Alginate-Specific Production Rate and Yields in Continuous Cultures

Figure 3 shows the alginate-specific production rate and the yields in the steady state under diazotrophic and nondiazotrophic conditions. Under diazotrophic conditions, a higher q$_p$ was obtained, with values that varied between 0.21 and 0.24 g g$^{-1}$ h$^{-1}$ (Figure 3a). It is clear that under diazotrophic conditions, a lower biomass concentration was reached (Figure 1a,c), which can explain the highest specific production rate obtained in the steady state conducted under diazotrophy. Previously, García et al. [32] described an increase in q$_p$ under ammonium limitation in chemostat cultures of *A. vinelandii* compared to cultures with an excess ammonium concentration at steady state.

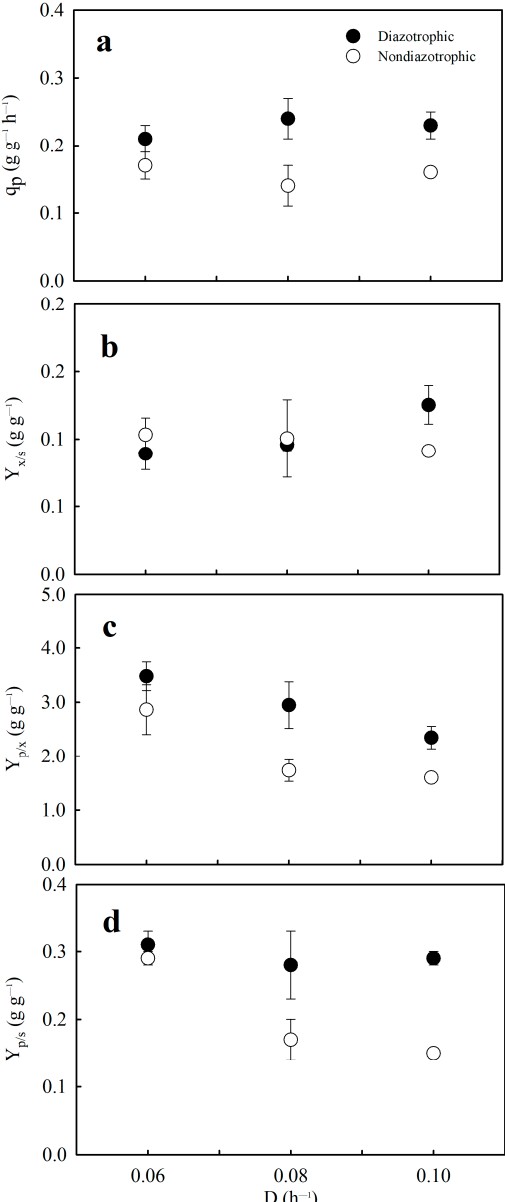

**Figure 3.** Alginate specific production rate ($q_p$) (**a**), yield of biomass based on sucrose ($Y_{x/s}$) (**b**), yield of alginate based on biomass ($Y_{p/x}$) (**c**), yield of alginate-based on sucrose ($Y_{p/s}$) (**d**) in continuous cultivations of *A. vinelandii* grown under diazotrophic (black circles) and nondiazotrophic conditions (white circles).

Similar $Y_{x/s}$ values were obtained under both conditions at a D of 0.06 and 0.08 h$^{-1}$ (Figure 3a). Under nondiazotrophic conditions, the $Y_{x/s}$ was similar at all the D values studied, reaching a value of approximately 0.10 g g$^{-1}$ (Figure 3b). However, under diazotrophic conditions, a higher $Y_{x/s}$ (0.13 ± 0.01 g g$^{-1}$) was observed when the cultures were conducted at a D of 0.10 h$^{-1}$ (Figure 3b). In this case, this increase in the $Y_{x/s}$ can be explained by a higher level of carbon diverted to the biomass (see below Table 3). Under diazotrophic conditions, the $Y_{p/x}$ decreased from 3.5 ± 0.3 to 2.3 ± 0.2 g g$^{-1}$ as D increased from 0.06 h$^{-1}$ to 0.10 h$^{-1}$ (Figure 3c). In cultures conducted at a D of 0.06 and 0.10 h$^{-1}$, the $Y_{p/x}$ was significantly higher under diazotrophic conditions, which could be associated with the protection of nitrogenase for nitrogen fixation, as previously described by Sabra et al. [11]. In this sense, it is possible that under diazotrophy the quantity of alginate by a unit of cells produced is increased as a mechanism to decrease the oxygen diffusion to the cells. Comparing the alginate yields reveals that the yield of alginate based on

biomass obtained in our study is about 25-fold higher than those reported from algae [33]. A possible reason for this difference could be related to the process of extraction of alginate algal, whereas the produced *A. vinelandii* cells are excreted, and depending on the growth conditions, its biosynthesis could be favored over biomass production.

**Table 3.** The carbon distribution in *A. vinelandii* continuous cultures under diazotrophic and nondiazotrophic conditions.

| Condition | Diazotrophy | | | Nondiazotrophy | | |
|---|---|---|---|---|---|---|
| D (h$^{-1}$) | 0.06 | 0.08 | 0.10 | 0.06 | 0.08 | 0.10 |
| Biomass (%C-mol) | $10.0 \pm 1.3$ | $10.7 \pm 0.5$ | $14.0 \pm 1.6$ | $11.7 \pm 1.6$ | $11.2 \pm 3.2$ | $10.1 \pm 0.1$ |
| Alginate (%C-mol) | $27.5 \pm 1.5$ | $25.3 \pm 4.8$ | $26.1 \pm 0.7$ | $26.8 \pm 0.1$ | $15.2 \pm 2.60$ | $13.0 \pm 0.2$ |
| P3HB (%C-mol) | $6.2 \pm 0.1$ | $1.2 \pm 0.3$ | $1.0 \pm 0.1$ | $1.8 \pm 0.3$ | $1.0 \pm 0.1$ | $1.2 \pm 0.4$ |
| $CO_2$ (%C-mol) | $58.9 \pm 3.1$ | $56.0 \pm 3.9$ | $51.5 \pm 4.9$ | $63.1 \pm 7.3$ | $57.3 \pm 17.2$ | $35.7 \pm 2.3$ |
| C-recovered (%mol) | 103 | 93 | 93 | 103 | 85 | 60 |

Similar to the evidence obtained for $q_p$ (Figure 3a), under diazotrophic conditions, $Y_{p/s}$ was not affected by changes in D, reaching a value of approximately 0.30 g g$^{-1}$ (Figure 3d). However, under nondiazotrophic conditions, the $Y_{p/s}$ decreased from 0.29 g g$^{-1}$ to 0.15 g g$^{-1}$ by increasing the D. Comparing the evidence obtained in the conditions evaluated indicates that under diazotrophy, a higher conversion of sucrose to alginate can possibly be obtained at D values of 0.08 and 0.10 h$^{-1}$ (Figure 3d). Based on this, the effect of D on the $Y_{p/s}$ could be related to changes in carbon flux that allow a higher proportion of sucrose to be diverted to the alginate. To the best of our knowledge, the evidence obtained in this study is the first to evaluate alginate production (yields and specific production rates) under diazotrophic and nondiazotrophic conditions in chemostat cultures limited by oxygen.

*3.4. Carbon Balance at Different Dilution Rates under Diazotrophic and Nondiazotrophic Conditions*

The carbon distribution (percentage of carbon atoms from sucrose converted to alginate, biomass, P3HB and $CO_2$) at each steady-state condition is shown in Table 3. A higher percentage of carbon diverted to biomass ($14.0 \pm 1.6\%$) was obtained at a D of 0.10 h$^{-1}$ under diazotrophic conditions. In all the other conditions evaluated, the carbon diverted to biomass only varied between 10 and 11% (Table 3). In agreement with this evidence, Inomura et al. [10] observed that the carbon flux destined for biomass formation is similar for both diazotrophic and nondiazotrophic conditions using mathematical modeling. Under diazotrophic conditions, the carbon diverted to alginate reached values between 25.3% and 27.5%. Nevertheless, under nondiazotrophic conditions, the carbon diverted to alginate decreased approximately twofold by changing the D from 0.06 to 0.10 h$^{-1}$ (Table 3). Independent of the conditions evaluated, more carbon was diverted to $CO_2$ (above 51% under diazotrophic conditions), which could be indicative of a carbon flux more active through the TCA cycle. This finding confirms that under the conditions assayed, it is necessary to operate the chemostat under diazotrophic conditions to improve the carbon into alginate.

Under diazotrophic conditions, the carbon balances were between 93–103%, which indicates that the cells utilized sucrose efficiently to produce biomass, alginate, and $CO_2$. Under nondiazotrophic conditions and D values of 0.08 and 0.10 h$^{-1}$, the carbon balance was close to 85% and 60%, respectively (Table 3), which is a clear indication that other

carbon products were produced. Therefore, a change in metabolism could improve the production of organic acids or amino acids excreted into the culture medium [7,34]. Based on this evidence, further research could be developed to evaluate the effect of nitrogen fixation on the metabolism of carbon in *A. vinelandii* cells.

*3.5. Alginate Molecular Weight and Gene Expression of alg8 and alg44 in Continuous Cultures*

The algal alginates have several problems concerning their production which may limit their use in many interesting contexts, especially in the pharmaceutical and chemical industries, where polymers with a very well-defined composition are required. In the cultivations of *A. vinelandii*, the G/M ratio, molecular weight, and acetylation degree might be manipulated through the manipulation of culture conditions during the bioprocess. Specifically, the molecular weight is one of the most important chemical characteristics of alginates because this characteristic determines the rheological properties of this polymer [35]. In the present study, independent of the D assayed, higher alginate MMW values were obtained under diazotrophic conditions, particularly for D values of 0.08 and 0.10 $h^{-1}$ (Figure 4a). The alginate MMW values obtained from the steady state under diazotrophic conditions showed a bell-shaped behavior, achieving a maximum value of $725 \pm 20$ kDa at D = 0.08 $h^{-1}$ (Figure 4a). Under nondiazotrophic conditions, no significant differences were found in the alginate MMW produced at 0.06 and 0.08 $h^{-1}$. The higher values of alginate MMW achieved under diazotrophic conditions compared to nondiazotrophic conditions might be related to the protection of the nitrogenase complex, as has been previously proposed [11,26]. Sabra et al. [11] observed that alginate plays an important role in the protection of nitrogenase because it can form a capsule around the cells, decreasing oxygen diffusion to the cell. Those authors concluded that the alginate quality, not quantity, is the most important characteristic for the protection of nitrogenase for the diazotrophic growth of *A. vinelandii* in terms of the MMW.

Considering that at D = 0.08 $h^{-1}$, a greater difference was found in the alginate MMW synthesized under the conditions studied, the expression of the *alg8* and *alg44* genes was evaluated (Figure 4b). Figure 4b clearly shows that the expression of both *alg44* and *alg8* was higher under diazotrophic conditions, in which the expression of the *alg8* and *alg44* genes was 2.2- and 1.8-fold higher. These data suggest that both *alg44* and *alg8* gene expression can be influenced by nitrogen fixation conditions. Therefore, in cultures of *Pseudomonas aeruginosa*, the expression of the operon of alginate biosynthesis is negatively regulated by the sigma factor RpoN when ammonium is present in the medium [36].

Similar evidence has been reported in batch and continuous cultures of *A. vinelandii* [7,37]. In chemostat cultures, higher *alg8* expression can be related to a higher alginate molecular weight [7], suggesting an explanation at the cellular level for the changes in molecular weight. In agreement, Flores et al. [37] evaluated the transcription of genes involved in alginate polymerization and depolymerization under controlled DOT in batch cultures of *A. vinelandii*. These authors found that in cultures in which a high-molecular-weight alginate (1200 kDa) was synthesized, the transcriptions of *alg8* and *alg44* was considerably greater compared to those in cultures conducted at 5% DOT. The higher expression of *alg8* and *alg44* observed under diazotrophic conditions could explain the highest molecular weight obtained when the cells grew diazotrophically. In this study, we have demonstrated the feasibility of enhancing alginate production and quality (molecular weight) under nitrogen-fixing conditions in a continuous mode of operation. Although at present there is no information on production costs of alginates from *A. vinelandii*, since there is no commercial production, a process like the one described in the present study could be competitive for medical applications. This being said, pharmaceutical grade alginates with defined composition (molecular weight and G/M profiles) can have costs about 480 USD per gram (NovaMatrix, web catalog 2023).

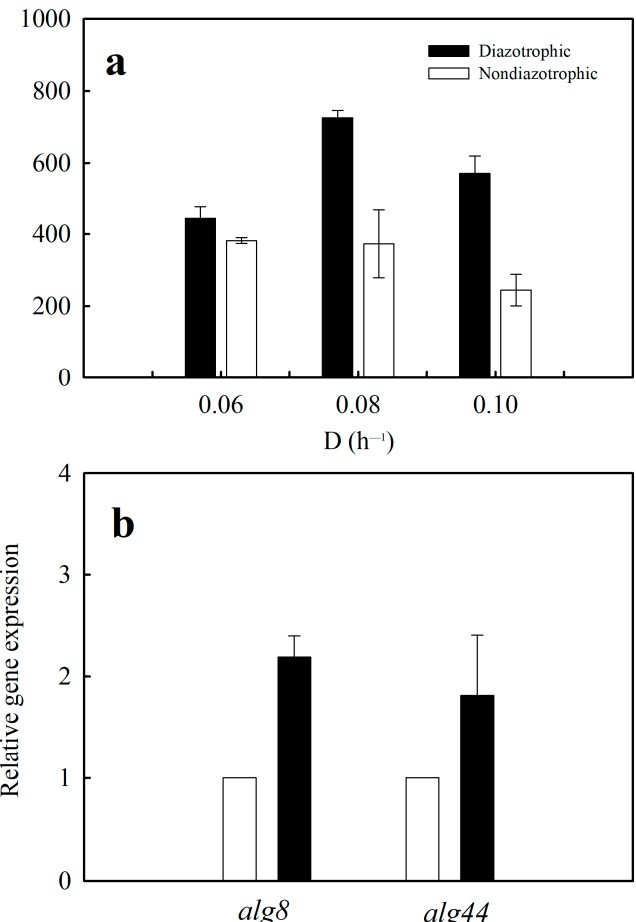

**Figure 4.** The mean molecular weight (MMW) of alginate a different dilution rate (**a**) and gene expression of *alg8* and *alg44* (**b**) in continuous cultures of *A. vinelandii* operated to D of 0.08 h$^{-1}$. Cultures under diazotrophic (black) and nondiazotrophic (white) conditions.

## 4. Conclusions

A comparative analysis of bacterial alginate production under diazotrophic and non-diazotrophic conditions was performed in a continuous bioprocess. A higher conversion of sucrose to alginate and alginate-specific production rates were obtained under diazotrophy (nitrogen-fixing condition). Under diazotrophic conditions the specific productivity was $0.24 \pm 0.03$ g g$^{-1}$ h$^{-1}$. A higher alginate molecular weight ($725 \pm 20$ kDa) was produced under diazotrophic conditions. Greater gene expression of *alg8* and *alg44* (encoding polymerases) can explain the higher molecular weight obtained. This study demonstrates that under nitrogen-fixing conditions, alginate production can be enhanced. The ability to maintain constant molecular weight in bacterial alginate production on batch cultures is recognized as a problem. The findings obtained in this study using continuous culture operating in a steady state indicate that this modality can be an appropriate way to control the characteristics of the alginate synthesized, and thus, this modality could be used for a potential industrial production of alginate under diazotrophic conditions.

**Author Contributions:** P.C.-A.: Investigation, Writing. V.U.: Investigation, Writing. B.P.: Investigation, Writing. T.C. and C.P.: Writing—review & editing. A.D.-B.: Conceptualization, Resources, Funding acquisition, Supervision, Writing—review & editing. All authors have read and agreed to the published version of the manuscript.

**Funding:** This research was funded by the Agencia Nacional de Investigación y Desarrollo (ANID) Chile: Beca de Doctorado Nacional ANID N°21201148 and Red ANID FOVI210018.

**Institutional Review Board Statement:** Not applicable.

**Informed Consent Statement:** Not applicable.

**Data Availability Statement:** All data will be provided to any interested part upon its requesting.

**Conflicts of Interest:** The authors declare no conflict of interest.

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
