# Peer review of "Continuous Bioproduction of Alginate Bacterial under Nitrogen Fixation and Nonfixation Conditions"

_fermentation, doi:10.3390/fermentation9050426_

Round 1
Reviewer 1 Report
Accepted
Thank You very much for the opportunity to review this manuscript.
Author Response
The reviewer does not include specific observations.
Reviewer 2 Report
I am very grateful you for the invitation to review manuscript fermentation-2366721 by Contreras-Abara and coauthors "Continuous bioproduction of alginate bacteria under nitrogen fixation and nonfixation conditions”. The aim of this study was to develop a continuous bioprocess and a comparative analysis of alginate production under diazotrophic and nondiazotrophic conditions. The work is interesting but needs adjustments to increase the quality of the material.
Comments:
- Abstract: Insert the alginate bioproduction mechanism.
- Lines 11-12: Applications and characteristics are different information.
- Abstract: Please indicate in the abstract a brief and better step-by-step about the work and the evaluated parameters.
- Line 23: Change the repeated keywords by different words from the title.
- Line 28: Insert other applications to justify the term “diverse”.
- Lines 41-43: Algae production can be considered a bioprocess.
- Line 58: check the correct citation of references.
- Introduction: Review the presentation of papers in the introduction. The details should be presented in the discussion. The introduction should be used to present general aspects.
- Introduction: Include a better explanation of diazotrophic and nondiazotrophic conditions.
- Lines 109, 123 and others: Please standardize the units (minutes, min, etc).
- Line 180: Remove the term “Figure 1” from the figure.
- Line 267: Remove the term “Figure 3” from the figure.
- Results: Include an improved discussion of yield and process versus conventional alginate sources.
- Results and discussion: The authors should improve the discussion of the advantage of using alginate production by the suggested process in relation to the conventional process.
- General comments: The demand for alginate is unclear. Include market information, quantity used in the most diverse processes, among other information.
- Discuss in relation to the cost of the process in relation to the conventional procurement process. Also, specify the conventional process for better comparison.
Reviewer 3 Report
Good experimental plan and methods applied. Discussions of the obtained results should be improved and conclusion reviewed.
I suggest minor revisions before publication.

Minor editing of English form is required.
